# Impact of Posttranslational Modification in Pathogenesis of Rheumatoid Arthritis: Focusing on Citrullination, Carbamylation, and Acetylation

**DOI:** 10.3390/ijms221910576

**Published:** 2021-09-30

**Authors:** Eui-Jong Kwon, Ji Hyeon Ju

**Affiliations:** 1Department of Medicine, Graduate School of Medical Science, The Catholic University of Korea, Seoul 06591, Korea; ejkwon@catholic.ac.kr; 2Chemical, Biological, Radiological, and Nuclear (CBRN) Defense Research Institute, Armed Forces CBRN Defense Command, Seoul 06591, Korea; 3Department of Internal Medicine, Division of Rheumatology, Seoul St. Mary’s Hospital, College of Medicine, The Catholic University of Korea, Seoul 06591, Korea; 4CiSTEM Laboratory, Catholic iPSC Research Center (CiRC), College of Medicine, The Catholic University of Korea, Seoul 06591, Korea; 5Department of Biomedicine & Health Science, Seoul St. Mary’s Hospital, College of Medicine, The Catholic University of Korea, Seoul 06591, Korea

**Keywords:** rheumatoid arthritis, pathogenesis, posttranslational modification (PTM), citrullination, carbamylation, acetylation, anti-modified protein antibodies (AMPAs), anti-citrullinated protein/peptide antibodies (ACPAs), anti-carbamylated protein (anti-CarP) antibodies, anti-acetylated protein antibodies (AAPAs)

## Abstract

Rheumatoid arthritis (RA) is caused by prolonged periodic interactions between genetic, environmental, and immunologic factors. Posttranslational modifications (PTMs) such as citrullination, carbamylation, and acetylation are correlated with the pathogenesis of RA. PTM and cell death mechanisms such as apoptosis, autophagy, NETosis, leukotoxic hypercitrullination (LTH), and necrosis are related to each other and induce autoantigenicity. Certain microbial infections, such as those caused by *P**orphyromonas*
*gingivalis*, *Aggregatibacter actinomycetemcomitans,* and *Prevotella copri*, can induce autoantigens in RA. Anti-modified protein antibodies (AMPA) containing anti-citrullinated protein/peptide antibodies (ACPAs), anti-carbamylated protein (anti-CarP) antibodies, and anti-acetylated protein antibodies (AAPAs) play a role in pathogenesis as well as in prediction, diagnosis, and prognosis. Interestingly, smoking is correlated with both PTMs and AMPAs in the development of RA. However, there is lack of evidence that smoking induces the generation of AMPAs.

## 1. Introduction

Rheumatoid arthritis (RA), the most common form of chronic inflammatory arthritis, is mainly targeting synovial joints [1,2,3]. However, RA is also a systemic autoimmune disease that involves not only joints but other organs such as the lungs, pericardium, sclera, peripheral nerves, skin, and vessels [4,5,6]. Untreated RA destroys the articular cartilage and nearby bones, resulting in functional disability [7,8]. The current strategy for RA treatment focuses on early and aggressive management before irreversible articular damage [7,9]. Thus, recent research has focused on events occurring before the presentation of RA; specifically, the pathogenesis and preclinical stage.

The 2010 American College of Rheumatology (ACR)–European League Against Rheumatism (EULAR) criteria are often used as the basis for a diagnosis of RA (Table 1). The new scoring system results in a score of 0–10, and a score ≥ 6 is considered satisfactory for the diagnosis of RA. The 2010 ACR–EULAR criteria include anti-citrullinated protein/peptide antibodies (ACPAs) and rheumatoid factor (RF). The diagnostic criteria for ACPAs are the presentation of an early disease course and the prediction of an aggressive disease course [10].

The pathogenesis of RA has not yet been fully identified, as the characteristic pathological features make it difficult to identify the causative factors [7,11,12,13]. First, RA is the result of the interaction between numerous genetic, environmental, and immunological factors. Second, the various backgrounds of races and ethnicities each have different trigger factors, which further complicate diagnosis. In addition, all the causative factors have usually been interacting for a prolonged duration before the onset of RA.

## 2. Pathogenesis

The average prevalence rate of RA is 0.1% to 1.0%, and the condition, which is more common in women than it is in men, has the highest rate among rheumatologic diseases [14]. However, the prevalence and occurrence rate vary according to race and ethnicity [15,16,17]. The differences exist not only in prevalence but also in disease activity and clinical outcomes [16,18]. There are also differences in the frequency of human leukocyte antigen (HLA) alleles, single-nucleotide polymorphisms (SNPs), and disease manifestation [19,20]. The most important genetic factor is shared epitope (SE) of HLA-DRB1 of major histocompatibility complex (MHC) [21,22,23]. The HLA-DRB1 allele associated with MHC is the most popular genetic factor in RA, and it increases the risk (HLA-DRB1*0401, *0404/*0408, *0405, *0101, *1001, and *1402), whereas HLA-DRB1*13 has a protective effect and decreases the risk [3,20,23,24,25,26].

However, other genetic factors also exist in non-HLA regions, such as peptidylarginine deiminase (*PAD*), signal transducers and activators of transcription 4 (*STAT4*), protein tyrosine phosphatase N22 (*PTPN22*), tumor necrosis factor (TNF) receptor-associated factor 1-C5 (*TRAF1-C5*), and interleukin (IL)-1 receptor-associated kinase 1 (*IRAK1*) genes [20,27,28,29,30,31]. Genes involved in T cell activation or the nuclear factor (NF)-κB pathway and SNPs are linked in RA [1,32].

Genome-wide association studies (GWAS) are widely used to identify gene candidates that correlate with RA [1,2,3,33,34]. Recently, a large Korean cohort study reported that the SLAMF6, CXCL13, SWAP70, NFKBIA, ZF-P36L1, and LINC00158 loci may be new genetic factors [35]. In a Chinese cohort, the potential involvement of the IL12RB2, BOLL-PLCL1, CCR2, TCF, and IQGAP1 loci were also identified through a GWAS study [36]. When environmental factors such as smoking, microorganisms, race, and periodontitis are combined with genetic factors, the immune tolerance breaks down [7,24,37,38]. For example, commencement of smoking by a person with the *HLA-DRB1 SE* gene in-creases the potential to develop RA [25,39].

Certain infectious microorganisms (e.g., Epstein–Barr virus (EBV), parvovirus B19, *Proteus* sp., and *Escherichia coli*) may have cross-reactivity for sensitizing autoantigens by molecular mimicry [40,41,42]. *Porphyromonas gingivalis*, the major pathogen of periodontal disease, expresses the bacterial *PAD* gene and leads to citrullination [43,44]. The gut microbe *Prevotella copri* accumulates in the feces and has homologous epitopes of N-acetylglucosamine-6-sulfatase (GNS) and filamin A (FLNA), which suggests that it is a potential RA trigger [45]. PAD changes the characteristics of proteins by citrullination and induces autoantigenicity [46,47,48]. Citrullination is followed by the production of ACPAs, the key molecule involved in RA pathology, in genetically susceptible individuals [49].

The normal diarthrodial joint space is filled with a small amount of synovial fluid (SF) [50], and the innermost part of the joint capsule and tendon sheath is covered with one to three layers of specialized cells, called the synovial membrane [51]. Synovial cells connect to each other through cadherin 11 [52]. The two types of synoviocytes are type A macrophage-like synoviocytes and type B fibroblast-like synoviocytes (FLS) [52,53]. RA progression with synovitis leads to edema of the synovium and an increase in the extracellular matrix. Abnormal proliferation structure of synovial tissue, called pannus, then forms [3,54].

The pannus infiltrates nearby structures, including the normal synovium, cartilage, and bony structure [55,56]. Synoviocytes in RA patients have abnormal proliferation traits similar to those of cancer cells [57], and mutations in *p53*, a tumor suppressor gene, have been reported in the synovium of these patients [58,59,60]. This phenomenon correlates with hyperactivated synoviocytes, which are apoptosis-resistant [61]. The variable mutation patterns of *p53* may correlate with the heterogeneity of RA [59].

Dysregulation of the cell death mechanism is a causative factor at any step of RA pathogenesis, and in RA, synoviocytes exhibit altered apoptotic responses [62]. Autophagy may be dysregulated in severe conditions, such as cell senescence or growth factor starvation, inducing self-cannibalism [46]. Any dysregulation of the cell death mechanism could be a source of autoantigens by introducing an epitope of intracellular molecules to the naïve immune system.

Neutrophil extracellular traps (NETs) are formed by inflammatory stimuli in a process called NETosis, which captures invading microorganisms [63]. NETosis is another form of cell death that is distinguishable from apoptosis and necrosis [64,65]. NETs are composed of extruded intracellular components such as DNA, histones, granular proteins (myeloperoxidase (MPO), elastase, and lactoferrin), and cytoplasmic proteins (such as calprotectin and catalase.) [46,63]. This suggests that NETosis leads to exposure of intracellular and intranuclear molecules, which can induce autoimmunogenicity [66].

Recent studies claim that citrullination is not a common pathway associated with NETosis [67]. Therefore, the concept of leukotoxic hypercitrullination (LTH) and defective mitophagy has emerged as a more precise description [67,68]. Although LTH is the main source of de novo pathogenic citrullination in RA, NETosis may act as a redistributor of steady-state citrullinome in neutrophils, also inducing autoimmunogenicity [68].

Any event causing the breakdown of self-tolerance triggers the antigen presenting cell (APC) to sense the autoantigen, such as fibrinogen, vimentin, enolase, and so on, which then expresses the antigen with the MHC II [69]. The antigen presented by the MHC II stimulates the T cell receptor (TCR), which is expressed on the surface of cluster of differentiation 4 positive (CD4+) T cells [69,70]. However, this process is not sufficient to activate T cells. The surface molecule CD80/86 of the APC also binds to CD28 on the T cell surface [71], and binding between CD80/86 and CD28 acts as a “co-signal” for T cell activation [72].

Activated T cells differentiate into helper T1 (T_H1_) and T17 (T_H17_) cells, which express CD40 ligand (CD40L) [73]. CD40L of CD4+ T_H_ cells binds to CD40 of B cells, inducing B cell differentiation into plasma cells, which secrete autoantibodies such as RF and ACPAs [74]. Autoantibodies bind to autoantigens to form immune complexes, which activate the complement system. Furthermore, T_H_ cells produce numerous cytokines such as interferon (IFN)-γ, TNF-α, lymphotoxin-β, ILs (especially IL-6 and IL-17), and granulocyte-macrophage colony-stimulating factor (GM-CSF). Rituximab targets CD20, sometimes called “medical splenectomy” and is used for RA treatment because it depletes B cells [75].

Several groups of T cells, called effector T cells (T_Eff_ cells), rapidly respond to these stimuli and activate macrophage-like synoviocytes and FLS [76]. Activated macrophage-like synoviocytes release proinflammatory cytokines including TNF-α and ILs (IL-1, IL-6, IL-12, IL-15, IL-18, and IL-23), whereas activated FLS produce IL-1, IL-6, and TNF-α [77]. Recent theory suggests that T_H17_ cells are more important than T_H1_ cells are in RA pathophysiology [78,79]. T_H17_ cells secrete IL-17, which enhances the upregulation of inflammatory cytokine production in synoviocytes, including TNF-α, IL-1, and IL-6. It also mediates neutrophil, granulopoiesis, and osteoclast differentiation. IL-23 from T_H17_ cells affects the activity and glycosylation of autoantibodies [80]. The T_H1_/T_H17_ ratio in patients with RA, is inversely proportional to disease activity [81].

The Janus kinase (JAK)/STAT signaling pathway in these inflammatory cells amplifies the immune response [82]. Other intracellular signaling pathways are also involved, such as spleen tyrosine kinase (Syk), mitogen-activated protein kinases (MAPKs), and NF-κB [83]. Inhibition of these pathway is one of the mechanisms of action of small-molecule drugs. JAK inhibitors such as tofacitinib, baricitinib, and upadacitinib have been approved for the treatment of RA by the US Food and Drug Administration (FDA) [84]. Fostamatinib, which blocks Syk, usually used in chronic immune thrombocytopenia (ITP), has been in trials for RA treatment [85,86,87].

Regulatory T (T_Reg_) cells expressing CD25 and the transcription factor forkhead box P3 (FOXP3) regulate other immune cells to maintain tolerance [88]. Some studies suggest that the loss of cytotoxic T lymphocyte antigen 4 (CTLA-4) expression interferes with the suppressive role of T_Reg_ cells [89]. Abatacept, a fusion protein including the constant fragment (Fc) region of immunoglobulin (Ig) G1 and the extracellular part of CTLA-4, competes with CD28 of T cells such as T_Reg_ cells. Because the binding affinity of abatacept to CD80/86 of APC is higher than that to CD28, abatacept modulates excessive immune responses in RA [72].

TNF-α stimulates the proliferation of T cells and B cells and activates FLS to produce matrix metalloproteinases (MMPs) [1,90]. Autophagy is upregulated in FLS after TNF-α exposure, and RA patients show higher levels of autophagy than normal persons do [91]. TNF-α also upregulates adhesion molecules on endothelial cells and their pathological neovascularization [1]. Production of IL-1, IL-6, and GM-CSF is accelerated by TNF-α [92], whereas IL-6 interacts with TNF-α to promote the cell cycle for the proliferation of FLS, leading to RA induction [93,94]. TNF-α induces the expression of dickkopf-1 (DKK-1), which downregulates the Wnt receptors of osteoblast precursors [94]. TNF-α also interacts with osteoclast precursors and osteoblasts (OBs), leading to pathological bone destruction in RA by fusing osteoclast precursors to form activated osteoclasts (OCs), stabilizing OB, inducing osteocyte apoptosis, and enhancing bone absorption [95,96,97]. Figure 1 shows a schematic illustration of RA pathogenesis.

Ai, R. et al. [98] found that there are epigenetically similar regions in the FLS of RA patients, including a study that found that Huntingtin-interacting protein-1 (HIP-1) in the Huntington’s disease signaling pathway is correlated with FLS in RA patients. The action of HIP-1 is related to FLS invasion of the matrix, which regulates the severity of RA [98,99].

Air pollutants appear to play a role in RA pathogenesis, and the ACPAs titer could be predicted by exposure to particulate matter with a diameter ≤2.5 μm (PM 2.5) [11]. In addition, ozone exposure and living near high-traffic roads were recognized as risk factors for RA in a meta-analysis [100]. A body mass index (BMI) indicating adiposity is linked to the risk of RA development, and this is more significant in women [13]. Some studies have shown that the gut microbiome and its metabolites may induce RA by stimulating T_H17_ cells of mucosal immune tissue that control the production of autoantibodies [101,102,103,104].

Disease-modifying antirheumatic drugs (DMARDs) are a group of drugs that regulate the activity of RA. In recent decades, biological DMARDs have been developed based on the pathophysiology of RA, particularly targeting certain molecules or pathways. Table 2 shows a list of biologic DMARDs targeting cytokines and cell-surface molecules approved by the FDA [84].

## 3. Citrullination in RA

Citrullination is a process that converts the amino acid arginine to citrulline [46], and it is catalyzed by a Ca^2+^-dependent enzyme, PAD [105,106]. Every event where citrulline is converted to arginine increases the mass by 0.984 Da and the loss of one positive charge [105]. This posttranslational modification (PTM) alters acidity, which affects the isoelectric point (pI), ability to form hydrogen bonds, interaction with other amino acid residues, and protein unfolding [107,108]. Furthermore, these changes can influence the function and half-life of the associated proteins [109], suggesting that citrullination may create a new protein [106,110]. The formation of a citrullinated protein also suggests the possibility of generating new epitopes that could act as a new autoantigen that escapes self-immune tolerance [111]. Citrullinated peptides have a higher binding affinity to the HLA-DRB1 (DRB1*0401 or *0404) antigen-binding groove than to the corresponding arginine-containing peptide [112].

The Ca^2+^ dependency of PAD, which catalyzes citrullination, may be the main switch for the regulation of citrullination in the body [68]. PAD requires not only Ca^2+^ but also a reducing environment to maintain free thiol cystine activity [113]. The oxidative environment of the extracellular space inhibits citrullination [114]. Furthermore, the body regulates the concentration of Ca^2+^ using numerous channels and hormones by investing energy.

Citrulline-specific CD4+ T cells have been found in both human and mouse models [115,116] and citrulline-specific T_H1_ and T_H17_ cells are increased in RA patients [117,118]. The sequence of human citrullinated enolase peptide-1 (CEP-1) is similar to that of the α-enolase of *P. gingivalis*; the anti-CEP-1 antibodies and the enolase of *P. gingivalis* also have cross-reactivity with equivalent epitopes [119]. Administration of glucocorticoids as an RA treatment may reduce the level of citrullination [120,121]. A significantly higher proportion of citrullinated protein has been detected in synovial biopsy samples from RA patients than in the synovium of healthy individuals [120].

In chronic obstructive pulmonary disease (COPD) patients and smokers, vimentin levels are increased [122]. COPD patients without RA sometimes show positive test results for anti-mutated citrullinated vimentin antibodies (anti-MCV), one of the ACPA tests, and anti-MCV positivity is correlated with the manifestation of a severe form of extra-articular RA [123]. Additionally, PAD2 expression accelerates citrullination in the lungs, which is enhanced by smoking [124,125]. In a rat model of autoimmune encephalitis, Odoardi et al. [126] demonstrated that the autoimmune cells acquire the capacity to enter the CNS only after residing within the lung tissue; first the autoimmune cells drained to lymphatics of airways, then entered to blood circulation to reach for the CNS. In conclusion, Valesini et al. [46] hypothesized that citrullination in the lung could be an extra-articular factor in the origin of RA autoimmunogenicity that generates lung-resident autoreactive T cells, which migrate to other target organs by the upregulation of chemokine receptors—as in case of a rat model of autoimmune encephalitis [126].

### 3.1. PAD Family

PAD, which hydrolyzes guanidinium side chains in peptidylarginine to peptidylcitrulline and ammonia [30], belongs to another larger group of enzymes called the amidinotransferase superfamily. Isoforms of PAD share approximately 50% sequence similarity [113]. PAD5, a designation not currently used, was once considered to be a human homolog of rodent PAD4; however, it was proved that the PAD5 is identical to PAD4, as indicated by expression, sequence data, and genomic organization [30]. Another notable type of PAD is found in eukaryotes. The PAD of *P. gingivalis* (PPAD), the major pathogen of periodontitis, is independent of the Ca^2+^ concentration [127,128], which makes it active at higher pH, and it has a preference for C-terminal arginine citrullination, regardless of whether it is the peptide-bound or free form [128,129]. When arginine gingipains get activated, bacterial enzymes similar to human trypsin, they cut polypeptides into short peptide fragments with a C-terminal arginine [46]. Then, PPAD rapidly citrullinates C-terminal arginine in the fragment [130]. Table 3 shows the site of expression and substrate of the five isoforms of PAD in addition to those of PPAD.

Autocitrullination is occasionally considered to mediate the inactivation and regulation of enzymes, but its definite role is still controversial [48,146,147]. PAD4 has multiple citrullination sites including the Arg-372 and Arg-374 of PAD, which are considered potential autocitrullination targets. Autocitrullination modifies the structure of PAD4 and may increase its recognition by autoantibodies [48]. Activity of PAD and PPAD is elevated in patients with both RA and periodontitis [147]. Citrullination by PPAD may induce an immunologic response against citrullinated proteins in RA patients with periodontitis and SE [46]. A higher IgG anti-*P. gingivalis* antibody titer is associated with HLA-DRB1 neutral alleles [148].

The roles of PAD2 and PAD4 have been identified in RA, and they have been detected in macrophages of the SF of RA patients (RA-SF) and granulocytes isolated from the synovium of a mouse arthritis model, respectively [149,150]. Autocitrullination of PAD4 enhances the chance of its recognition by autoantibodies, and anti-PAD4 antibodies have predictive and prognostic value in RA [48,151]. PPAD also undergoes autocitrullination and generates antibodies that are cross-reactive with other citrullinated proteins of the human body [128]. However, in contrast with anti-PAD4 antibodies, anti-PPAD antibodies have no correlation with disease activity or ACPAs levels [142].

### 3.2. ACPAs, a Diagnostic and Prognostic Tool in RA

ACPAs are a group of antibodies that sense citrulline-containing proteins/peptides and share partial cross-reactivity [152,153]. An in vitro study showed that 66% of ACPAs showed cross-reactivity with different epitopes, whereas 33% were mono-active [154]. ACPA is a well-established diagnostic serology test molecule for RA, with a specificity of 85–95% and a sensitivity of 67% [155,156,157,158]. The specificity and sensitivity of the ACPA test has improved gradually from the first generation (anti-CCP1 test) to the third generation (anti-CCP3 test) [46,159]. ACPA is useful for the prediction of RA because it already exists before the onset of RA [160], and its positivity indicates a more erosive disease course and severity [161,162,163].

Although there are various isotypes of ACPAs, such as IgG, IgA, and IgM, the IgG isotype is the most dominant form in RA patients [164]. The Fc region of ACPA undergoes remodeling, which causes its Fc fragment to show two characteristics: decreased galactosylation and sialylation and increased core fucosylation, which differs from those of other serum antibodies [165,166,167,168,169]. Sialylation has a protective effect against the autoimmunogenicity of anti-type II collagen antibodies [165].

Therefore, remodeling of the Fc region could indicate the alteration of the functional activities of ACPA [165,168], and this remodeling of ACPA occurs before the change in total serum Ig [167,170]. In vitro, ACPA binds to the Fc receptors of myeloid lineage immune cells, activating the component system through both classical and alternative pathways [171,172]. Citrullinated fibrinogen–ACPA complexes in RA patients could activate macrophages to release TNF-α [173]. IgM–RF enhances this cascade and extends the spectrum by inducing the secretion of other cytokines (IL-1β, IL-6, and IL-8) [171].

The Fc-glycan profile showed different Fc receptor and complement binding affinities [168]. The agalatosylated profile of the Fc of ACPA facilitates the production of high-affinity RF for agalactosylated IgG [166]. ACPA also undergoes extensive variable domain glycosylation associated with the SE allele, except for IgM ACPA [174,175]. The incorporation of N-linked glycosylation sites modulates the affinity of ACPA [166,168,176]. Because ACPAs are a collection of heterogeneous antibodies, the specificity of ACPAs differs even in individual patients [177,178]. Candidate citrullinated autoantigens are listed in Table 4.

The concentration of IgM–ACPA in the RA-SF, corrected for the total amount of IgG, was higher than that in the serum of the same patient, suggesting that there is local production of ACPAs [202]. Following binding to peripheral blood mononuclear cells, ACPA activates extracellular signal-regulated kinase (ERK) 1/2 and c-Jun N-terminal kinase (JNK) signaling pathways. This effect further enhances inhibitor of NF-κB kinase (IKK)-α phosphorylation, followed by activation of NF-κB and production of TNF-α [203]. Certain ACPAs, such as anti-citrullinated vimentin antibodies, induce NET formation, and NETosis provides the citrullinated autoantigen and PAD enzymes, which provides a positive feedback loop, further enhancing the formation of ACPAs [183].

In summary, ACPA has a pathogenic effect on RA, both in vitro and in vivo. First, ACPA induces NETosis, creating positive feedback. Second, ACPA stimulates macrophages to induce TNF-α and activate the complement of both classical and alternative pathways. Third, ACPA interacts with other autoantibodies associated with RA to initiate disease cascades. Fourth, these substances pre-existed before the onset of RA, called the pre-clinical stage, and their titer increases with a widening spectrum and stronger affinity during the development of RA. Fifth, the characteristics of ACPA could be altered through remodeling by PTM. Finally, these mechanisms mediate the applicability and usefulness of ACPA not only in the diagnosis of RA but also as an indicator of a more severe clinical course and structural damage.

A recent study by Chirivi et al. [204] showed that therapeutically inhibiting NET formation with ACPA, especially targeting the N-termini of histones 2A and 4, suppressed NET release or uptake by macrophages in various mouse models. Won et al. [205] detected circulating citrullinated antigens, such as type II collagen and filaggrin, in the sera of RA patients, including seronegative RA, using a monoclonal 12G1 antibody and proposed the possibility of it serving as a diagnostic tool for seronegative RA.

### 3.3. Citrullination and Cell Death Mechanism

Citrullination is the process of preparing for apoptosis, and it involves the induction of a functional loss of filament proteins [106]. In apoptosis, unlimited Ca^2+^ influx into the intracellular space activates the PAD, leading to citrullination [46]. In the normal apoptosis process, phagocytosis rapidly eliminates apoptotic bodies [206]. However, once dysregulation occurs during apoptosis or phagocytosis to eliminate apoptotic bodies, citrullinated intracellular antigens are exposed [207]. Similarly, the necrosis of neutrophils releases PAD into the extracellular space and leads to citrullination [208]. Compared with the SF of osteoarthritis (OA) patients, that of RA patients showed an elevation in the concentration of intranuclear materials and the activity of PAD4 [151]. PAD4 activity in vitro, released to the extracellular space, is higher in necrosis than it is in NETosis [209].

The names of newly identified cell death mechanisms, LTH and defective mitophagy, were formulated to distinguish them from other forms of neutrophil death that do not involve hypercitrullination, even if they form NET-like structures [68]. LTH releases citrullinated intranuclear molecules [210]. In LTH, perforin and the membrane attack complex (MAC), the pore-forming cytolytic proteins, induce the influx of Ca^2+^ and other ions, leading to osmotic lysis [211,212,213]. LTH is independent of ERK and nicotinamide adenine dinucleotide phosphate (NADPH) oxidase activity, a hallmark of NETosis [214,215].

Initiation of LTH originates not only endogenously but also exogenously. *Streptomyces* sp. have bacterial calcium ionophores, such as ionomycin and calcimycin [67]. Other bacterial pore-forming toxins also trigger LTH. The periodontal pathogen *Aggregatibacter actinomycetemcomitans* secretes the pore-forming protein leukotoxin A (LtxA) to activate citrullination in neutrophils [216,217,218].

### 3.4. Microorganisms Inducing Citrullination

*P. gingivalis* induces citrullination using PPAD independent of Ca^2+^ and pH level [43]. As described above, *A. actinomycetemcomitans* and *Streptomyces* sp. induce LTH that involves citrullination; *P. copri* and EBV have homologous epitopes that could trigger the production of ACPAs and autoimmunogenicity. Some other bacterial pathogens also induce membranolytic damage, which may induce citrullination in neutrophils and ACPA production via cell death mechanisms. Certain microbiomes in the lung, gut, and urothelium also produce pore-forming toxins that target neutrophils [219]. *Staphylococcus aureus* and *Streptococcus pyogenes* are microorganisms that colonize the extra-articular mucosa and produce Panton-Valentine leukocidin (PVL) and streptolysin O (SLO), respectively, which could induce the citrullination of neutrophils [67,220,221,222].

## 4. Carbamylation in RA

Carbamylation occurs in the human body under uremic or inflammatory conditions [223,224]. Carbamylation is both an enzymatic and nonenzymatic process of adding the “carbamoyl” part (−CONH_2_−), which is related to cyanate (^−^N=C=O), to proteins, or to peptides [225,226,227]. This PTM process leads to the production of homocitrulline or α-carbamyl-protein [226]. Carbamylation is thought to be an enzyme-independent process that typically occurs under uremic conditions [228,229]. Urea dissolved in water spontaneously generates cyanate, which is a carbamylate protein or peptide [230]. However, any inflammation causes oxidation of thiocyanate to produce cyanate, catalyzed by MPO and peroxide, regardless of whether the conditions are uremic or not [231].

Smoking may also induce carbamylation [232], and mononuclear cells of treatment-naïve RA patients show a correlation between autophagy and the level of carbamylation [233]. The carbamylated form of hemoglobin and low-density lipoprotein (LDL), detectable by laboratory tests, is correlated with acute or chronic renal failure and atherosclerosis, respectively [234,235,236,237,238]. Despite these findings, the precise role and effect of carbamylation in RA have not yet been definitively established.

### 4.1. Anti-Carbamylated Protein Antibodies, a Novel Hallmark for RA

Homocitrulline, which is generated by carbamylation, shows immunogenicity in RA, producing anti-carbamylated protein (anti-CarP) antibodies [239]. Anti-CarP antibodies are detected in RA patients, regardless of ACPA-positivity or ACPA-negativity [240]. Interestingly, anti-CarP antibodies and ACPA demonstrated definitive cross-reactivity in vitro [240]. The sensitivity and specificity of anti-CarP antibodies for RA diagnosis are 44% and 89%, respectively [241].

HLA-DR3 alleles, found at higher levels in ACPA-negative RA patients than in controls, are related to anti-CarP antibody-positive RA without ACPAs [242,243,244,245]. Although IgG levels of anti-CarP antibodies were shown to increase under uremic conditions and smoking in a mouse model, the levels did not increase in heavy smokers or show any association with smoking, or the association became insignificant after correcting for ACPAs in humans [244,246,247,248]. In ACPA-negative RA patients, positivity of anti-CarP antibodies is associated with more erosive manifestation of RA than negativity, independent of RF or ACPA [249,250,251]. Anti-CarP antibodies are also used to screen for those at risk, and the odds ratio is highest when all three autoantibodies (RF, ACPAs, and anti-CarP antibodies) are co-analyzed [252]. The combined presence of ACPAs and anti-CarP antibodies could strongly indicate RA [253].

### 4.2. Similarity between Anti-CarP Antibodies and ACPAs in RA

Studies have shown that carbamylation plays a role that is like citrullination in RA [249,254,255]. Similar to RF and ACPAs, anti-CarP antibodies show an association with risk when combined with smoking [256]. The presence of RF is associated with positivity for both ACPAs and anti-CarP antibodies [257]. Smoking induces both citrullination and carbamylation. However, there is a lack of evidence that smoking induces the production of autoantibodies; conversely, it correlates with the initiation of intolerance to multiple autoantigens, which may correlate with the overlapping of RF, ACPAs, and anti-CarP antibodies [258]. Specifically, cigarette smoking only induces autoimmunity, which is affected by HLA-DRB1 SE, and evidence does not support the notion that it induces de novo intolerance to citrullinated or carbamylated proteins or peptides [39].

Like ACPAs, anti-CarP antibodies could exist for several years before the onset of RA, and they could increase gradually in quantity just before disease presentation [254,259]. Notably, the presence of both ACPAs and anti-CarP antibodies strongly suggests RA, even if they are not only specific for RA [248]. The presence of each autoantibody is a marker of aggressive joint destruction in RA patients, making it possible to predict progression to RA in arthralgia patients [49,260]. Avidity of ACPAs and anti-CarP antibodies are lower than other antibodies to usual recall antigen; nevertheless, ACPAs and anti-CarP antibodies, composed of broad isotypes and subclasses (such as IgM, IgA, and IgG subclasses), undergo isotype-switching to diminish their avidity more [261].

## 5. Citrullination and Carbamylation in Other Disease

Published data suggest that smoking and periodontitis due to *P. gingivalis* are risk factors for cardiovascular disease [262]. Citrullinated proteins and PAD4 were detected in the atherosclerotic plaques in individuals without RA [263], and the ACPAs in RA patients can target these proteins from atherosclerotic plaques [264]. According to Hermans et al. [265], ST-segment elevation myocardial infarction (STEMI) is associated with ACPA. Citrullination is also associated with neurodegenerative diseases, cancers [105,144], and type 1 diabetes [266]. Citrullination of MBP in the CNS is correlated with the onset of MS [267,268], and it causes other autoimmune diseases, such as systemic lupus erythematosus (SLE) and autoimmune encephalomyelitis [268,269]. Because PAD and citrullination are involved in transcription, NET formation, and cell signaling, the connection between malignancy and citrullination has also been suggested [134,144].

Carbamylation is related to cardiovascular diseases [264] and plays a role as a pro-atherosclerotic activator by accumulating cholesterol in macrophages, inducing endothelial apoptosis, and increasing scavenger receptor recognition [231]. Cataracts are induced by carbamylation of α-crystallin [227]. Usually, the carbamylation of protein hormones downregulates the function of hormones; however, carbamylation of uncommon residues occasionally exhibits different effects on the intensity of downregulation, such as the A-chain of insulin, oxytocin, and arginine-vasopressin [270,271,272].

In the uremic status of end-stage renal disease (ESRD), the lysine residue of erythropoietin (EPO) is carbamylated and decreases in activity, resulting in the production of non-functional EPO, which leads to hypoxemia [273,274]. Patients with inflammatory bowel disease (IBD) showed significant differences in the proportion that presented with serum anti-CarP antibodies compared with normal controls [248]. Carbamylated LDL (c-LDL) shows low affinity for its receptor, which decreases the clearance rate in rabbits [275,276]. c-LDL induces the accumulation of cholesterol and the formation of foam cells and signals for inflammation [277], and it facilitates monocyte adhesion to the endothelium and vascular smooth muscle to enhance their proliferation [278].

## 6. Acetylation, Another Autoantibody-Inducing PTM Process

Acetylation, which normally occurs in the human body through both co-translation and post-translation [279,280], has two distinct pathways through which an acetyl group donated by acetyl-coenzyme A is attached to either the N-terminus of proteins or lysine residues [281]. Acetylation is a reversible enzymatic process catalyzed by various N-terminal and lysine acetyltransferases [281,282]. Similar to carbamylation, acetylation can modify lysine residues, change the molecular features of proteins or peptides, and induce the production of modified epitopes [160].

In addition, acetylated proteins can induce the production of anti-acetylated protein antibodies (AAPAs) [160]. In seronegative RA patients, acetylation of histones, a PAD-independent process, showed cross-reactivity with ACPA [283]. Kampstra et al. [284] showed that autoantibodies to PTM molecules, called anti-modified protein antibodies (AMPAs), which include ACPAs, anti-CarP antibodies, and AAPAs, could be part of a single concept because they share partial cross-reactivity with each post-translated autoantigen. Su et al. [285] showed that the dysregulation of acetylation attenuates the development of RA by downregulating FOXP3 expression.

## 7. Conclusions

Figure 2 shows a schema that illustrates posttranslational modifications (PTMs) and anti-modified protein antibodies (AMPAs) in RA pathogenesis. Citrullination and carbamylation are reported in a wide range of inflammatory tissues and, therefore, are considered as markers of inflammation rather than specific disease-dependent processes. PAD, which mediates citrullination, has five isoenzymes, including some with the potential to autocitrullinate. PPAD is an exogenous PAD independent of Ca^2+^ and pH. Acetylation is also involved in autoimmunogenicity and the production of autoantibodies in RA. Both carbamylation and acetylation have a common target of lysine residues, but the results of each PTM have distinguishable characteristics.

PTMs, including citrullination, carbamylation, and acetylation alone, are not sufficient to induce intolerance and autoimmunity, but this does not negate their importance. Because RA is caused by complex interactions between multiple pathogenic factors, the role of these PTMs is as significant as that of other pathogenic factors. PTMs are also correlated with one or more pulmonary, cardiovascular, neurodegenerative, malignant, or other autoimmune diseases, other than RA.

AMPA is closely related to the pathogenesis, prediction, diagnosis, and prognosis of RA, whereas smoking induces PTMs and interacts with AMPA to enhance the risk of RA development. Cell death mechanisms, including apoptosis, autophagy, NETosis, LTH, and necrosis induce or present PTM autoantigens, generating AMPAs. Some microbes such as *P. gingivalis* can cause self-citrullination of molecules to produce de novo epitopes, and *A. actinomycetemcomitans* uses LtxA to induce LTH in citrullinated neutrophils.

## Figures and Tables

**Figure 1 ijms-22-10576-f001:**
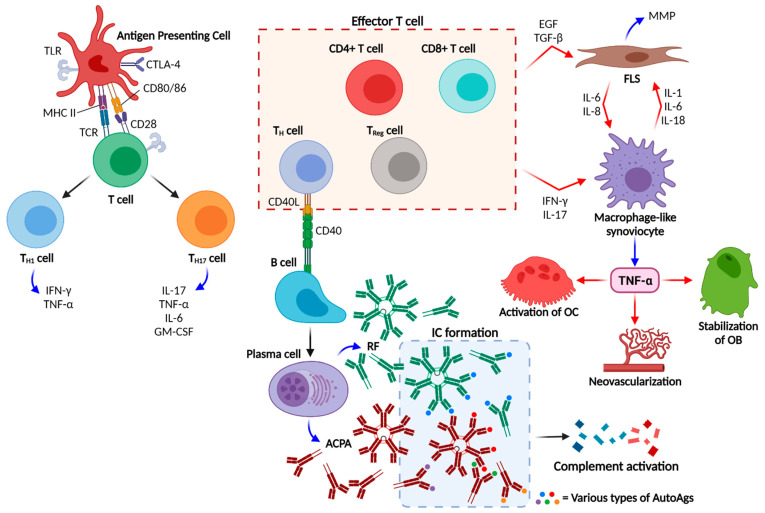
Schema of rheumatoid arthritis (RA) pathogenesis. Black, blue, and red arrows indicate differentiation, secretion of proinflammatory molecules, and stimulation, respectively. Broken red arrow indicates stimuli mediated by specific molecules. Abbreviations: TLR, Toll-like receptor; CD, cluster of differentiation; CTLA-4, cytotoxic T lymphocyte antigen-4; MHC, major histocompatibility complex; TCR, T cell receptor; T_H_ cell, helper T cell; T_H1_ cell, helper TH1 cell; T_H17_ cell, helper TH17 cell; T_Reg_ cell, regulatory T cell; RF, rheumatoid factor; ACPAs, anticitrullinated protein/peptide antibodies; IC, immune complex; AutoAg, autoantigen; EGF, epidermal growth factor; TGF-β, transforming growth factor-β; IFN-γ, interferon-γ; IL, interleukin; MMP, matrix metalloproteinase; FLS, fibroblast-like synoviocytes; OC, osteoclast; OB, osteoblasts [3]. Created with Biorender.com.

**Figure 2 ijms-22-10576-f002:**
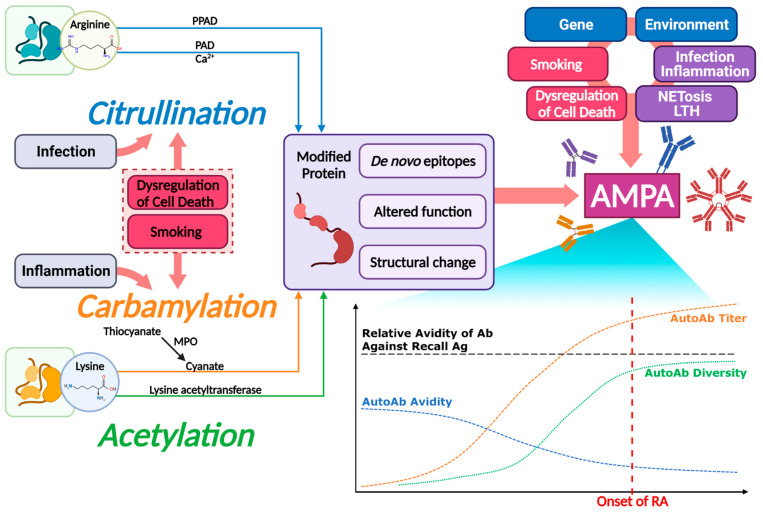
Schematic illustration of roles of posttranslational modifications (PTMs) and anti-modified protein antibodies (AMPAs) in rheumatoid arthritis (RA) pathogenesis. Citrullination, carbamylation, and acetylation self-modify proteins to generate autoantigenicity. Autoantigens induce the production of AMPAs. In pre-clinical stage of RA, the production of AMPAs is accompanied by the remodeling of antibodies. Simultaneously, AMPAs interact with other pathogenic factors of RA and undergoes isotype switching. Remodeling of autoantibodies lower the avidity of AMPAs, paradoxically heightening autoimmunogenicity. Titer and diversity of AMPAs gradually increase, driving the development of RA. All values in the graph are qualitative. Abbreviations: PAD, peptidylarginine deiminase; PPAD, PAD of *Porphyromonas gingivalis*; MPO, myeloperoxidase; LTH, leukotoxic hypercitrullination; Ab, antibody; Ag, antigen; AutoAb, autoantibody. Created with Biorender.com.

**Table 1 ijms-22-10576-t001:** Scoring for rheumatoid arthritis (RA) diagnosis.

Domain	Category	Score
Joint involvement	1 large joint (Shoulder, elbow, hip, knee, ankle)	0
	2–10 large joints	1
	1–3 small joints (MCP, PIP, thumb IP, MTP, wrists)	2
	4–10 small joints	3
	More than 10 joints (Including at least 1 small joint)	5
Serologic study	Negative RF and negative ACPAs (Under ULN)	0
	Low-positive ^1^ RF or low-positive ACPAs	2
	High-positive ^2^ RF or high positive ACPAs	3
Acute phase reactants	Normal CRP and normal ESR	0
	Abnormal CRP or abnormal ESR	1
Duration of symptoms	<6 weeks	0
	≥6 weeks	1

Revised 2010 ACR–EULAR criteria consist of four domains: joint involvement, serologic study including RF and ACPAs, acute phase reactants (CRP and ESR), and duration of symptoms [10]. ^1^ Low-positive, ≤3 × ULN ^2^ High-positive, ≥3 × ULN. Abbreviations: MCP, metacarpophalangeal joint; PIP, proximal interphalangeal joint; IP, interphalangeal joint; MTP, metatarsophalangeal joint; RF, rheumatoid factor; ACPAs, anti-citrullinated protein/peptide antibodies; CRP, C-reactive protein; ESR, erythrocyte sedimentation rate; ULN, upper limit of normal.

**Table 2 ijms-22-10576-t002:** Biologic and targeted synthetic disease-modifying antirheumatic drugs (DMARDs) for rheumatoid arthritis (RA).

DMARDs	Mechanism	Route of Administration
Abatacept	Fusion protein consists of extracellular domain of CTLA-4 and Fc region of IgG1, binding to CD80/86	IV, SC
Adalimumab	TNF-α inhibitor	SC
Infliximab		IV
Certolizumab pegol		SC
Golimumab		SC
Etanercept		SC
Rituximab	Monoclonal antibody against CD20	IV
Tocilizumab	Humanized monoclonal antibody against IL-6 receptor	IV, SC
Sarilumab		SC
Tofacitinib	JAK1/JAK3 inhibitor	PO
Baricitinib	JAK1/JAK2 inhibitor	PO
Upadacitinib	JAK1 inhibitor	PO
Filgotinib		PO
Peficitinib	Pan-JAK(JAK1/JAK2/JAK3/Tyk2) inhibitor	PO

Abbreviations: DMARDs, disease-modifying antirheumatic drugs; CTLA-4, cytotoxic T lymphocyte-associated protein 4; Fc, constant fragment; IgG, immunoglobulin G; CD, cluster of differentiation; TNF-α, tumor necrosis factor-α; IL, interleukin; JAK, Janus kinase; Tyk, tyrosine kinase; SC, subcutaneous; IV, intravenous; PO, per os (orally).

**Table 3 ijms-22-10576-t003:** Isoforms of peptidylarginine deiminase (PAD) compared with PAD of *Porphyromonas gingivalis* (PPAD).

Type	Site of Expression	Substrate	Reference
PAD1	Epidermis Uterus Hair follicle, sweat gland Stomach	Filaggrin Keratin-associated protein MEK1	[131,132]
PAD2	CNS, PNS Skeletal muscle Immune cells, spleen Skin, secretory gland Uterus, pancreas, kidney, inner ear	Myelin basic protein in CNS Histones Actin Vimentin RNAP2 β-catenin Enolase Fibrinogen	[133,134,135,136]
PAD3	Hair follicle Keratinocyte Peripheral nerve	Vimentin Filaggrin Trichohyalin Apoptosis-inducing factors	[137]
PAD4	Immune cells, spleen Secretory gland Brain Uterus Joints	Histones and nucleophosmin/B23 Type I collagen ING4 p300, p21, p53 Lamin C GSK3β PAD4	[138,139]
PAD6	Egg, ovary, testis, early embryo	Uncertain	[30,140]
PPAD	Enzyme of *P. gingivalis*	Fibrinogen α-enolase Collagen type II PPAD	[44,119,141,142]

There are five PAD isoenzymes. PAD2 stabilized by the NTZ, substrates β-catenin. Trichohyalin is a major structural protein in hair follicles. Note that PAD4 has a homodimer structure, the only type of PAD localized to the cell nucleus. PAD4 and PPAD can autocitrulate to generate antibodies. PAD6 is uniquely expressed in germ cells, and its precise role is unclear; it is thought to be essential for germ cell-specific structures in zygote/embryo development. PPAD is independent of Ca^2+^ concentration and active at higher pH; it prefers C-terminal arginine citrullination. Abbreviations: PAD, peptidylarginine deiminase; CNS, central nervous system; PNS, peripheral nervous system; PPAD, PAD of *Porphyromonas gingivalis*; NTZ, nitazoxanide [30,46,68,105,143,144,145].

**Table 4 ijms-22-10576-t004:** Candidate proteins and peptides as targets of anti-citrullinated protein/peptide antibodies (ACPAs).

Origin	Candidate Protein/Peptide	Site of Expression in Human	Reference
Endogenous	Fibrinogen	Inflamed joint	[179,180,181]
	Vimentin	Inflamed joint	[180,182,183,184]
	α-enolase	Joint tissue, inflammatory cells ^¶^	[180,181,185,186]
	Fibronectin	Plasma, synovium, SF	[187,188,189]
	Type II collagen	Joint tissue	[181]
	Histone ^§^	Nucleus	[190,191]
	BIP ^§^	ER	[170,192]
	Tenascin-C	ECM of joint	[193,194]
	Filaggrin	Epithelium	[195,196]
	Apo E	Plasma, CNS, RA-SF	[197,198]
	MNDA	Inflammatory cells ^¶^	[197,199]
	β-actin	Inflammatory cells ^¶^	[197,199]
	FUSE-BP ^§^	Nucleus	[200]
	hnRNP ^§^	Nucleus, RA-SF	[201]
Exogenous	Viral citrullinated peptides from EBV antigen	—	[190]
	α-enolase from *P. gingivalis*		[190]
	GNS sequence homologue of surface proteins of the *P. copri*		[45]

Fibrinogen, vimentin, and α-enolase are well-known targets of ACPAs. Numerous proteins and peptides are targets of ACPAs. Target proteins or peptides could originate from both endogenous and exogenous proteins. The site of expression did not consider the circulating autoantigen. ^§^ Ubiquitous expression pattern. ^¶^ Not limited to mentioned tissue only (usually nonspecific). Abbreviations: BIP, immunoglobulin binding protein; Apo E, apolipoprotein E; MNDA, myeloid nuclear differentiation antigen; FUSE-BP, far upstream element-binding proteins; EBV, Epstein–Barr virus; GNS, N-acetylglucosamine-6-sulfatase; SF, synovial fluid; ER, endoplasmic reticulum; ECM, extracellular matrix; RA-SF, synovial fluid of rheumatoid arthritis patient; CNS, central nervous system.

## Data Availability

Not applicable.

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
