# Peer review of "Impact of Posttranslational Modification in Pathogenesis of Rheumatoid Arthritis: Focusing on Citrullination, Carbamylation, and Acetylation"

_ijms, 2021, doi:10.3390/ijms221910576_

Round 1
Reviewer 1 Report
The review prepared by Kwon and Ju deals with the role of three posttranslational modifications (citrullination, carbamylation and acetylation) in the pathogenesis of rheumatoid arthritis. Although it discusses the mentioned posttranslational modifications in details, I have some remarks that need to be taken into account before its publication.
1. The title misleads the readers that the review looks at more posttranslational modifications associated with RA (for example hydroxylation, glycosylation, deamidation, glycation, chlorination etc.) , but there are only citrullination, carbamylation and acetylation. Although the reviewed modifications are the most studied recently, I suggest a correction of the title, for example:
“Impact of Citrullination, Carbamylation and Acetylation in the Pathogenesis of Rheumatoid Arthritis”
2. In the Introduction, the main target structure of RA (joints) is somehow neglected, emphasizing the other organs involved (lines 34-36), although the following sentences and the criteria in Table 1 show precisely the effects on this structure. However, it should be clarified that the main target structures are the joints.
3. Lines 131-146: The activation of T- and B-cells is presented as it is in the textbooks, without specifying and discussing the possible (supposed) autoantigens for RA. At least the candidate-antigens should be mentioned. The same is valid for Figure 1, where the antigen (autoantigen) is not even marked. Antigens are underestimated, but they are crucial for the pathogenesis in RA and for the subsequent activation of all given pathways.
4. Through the whole review is talking for posttranslational modifications of proteins/peptides without mentioning which are the proteins/peptides and in which tissue (structure) they are presented.
For example lines 238-239. From the articles is clear that this applies to fibrinogen, vimentin, enolase or synthetic peptides that are part of these proteins.
Minor notes:
Line 267: What means “… bacterial enzymes such as human trypsin, …”?
Lines 269-270: “Table 3 shows the expression and functions of the five isoforms of PAD in addition to those of PPAD.”
The functions are not presented in the Table.
Lines 294-295: What means “… other citrullinated proteins of human peptides”?
Line 322: “agalatocylated” should be corrected to “agalactosylated”
Author Response
All of our respond is in attached file.
Thank you for your efforts and comments.

Reviewer 2 Report
In this review, the Authors summarize the role of some PTMs (the best known) in RA pathogenesis. The work is well written but, in my opinion, it lacks of some information.
Since the PTMs considered in this review have been extensively described, in order to contribute to the field with new insights, I recommend to consider also others PTMs, which have recently been characterized in RA. Homocysteinylation, for example: different studies analyzed and/or identified antibodies directed to homocysteinylated proteins (Scand J Rheumatol. 2014;43(1):17-21. doi: 10.3109/03009742.2013.811537. Antibodies to N-homocysteinylated albumin and haemoglobin in patients with rheumatoid arthritis: a potential new marker of disease severity. Nowakowska-Plaza A. et al.; J Autoimmun. 2020;113:102470. doi: 10.1016/j.jaut.2020.102470. Homocysteinylated alpha 1 antitrypsin as an antigenic target of autoantibodies in seronegative rheumatoid arthritis patients. Colasanti T et al.). In my opinion, since the manuscript considered the “Impact of Posttranslational Modification in Pathogenesis of Rheumatoid Arthritis”, as it is in the title, at least this PTM should be also discussed.
In the paragraph “Pathogenesis”, the Authors referred “PAD changes the characteristics of proteins by citrullination and induces autoantigenicity. Citrullination is followed by the production of ACPA, the key molecule involved in RA pathology”. For sake of completeness and correctness, it should be added “Citrullination is followed by the production of ACPA, the key molecule involved in RA pathology, in genetically susceptible individuals”.
In the paragraph “Citrullination in RA”, the Authors reported that lungs are described as an “extra-site” for the autoimmunity beginning. Regarding this, could the Authors add some more literature works?
Author Response

(The authors gave the same response as above.)

Reviewer 3 Report
The authors summarized the understandings of post-translational modification in the pathogenesis of RA.
The manuscript has covered the topics of posttranslational modification in RA appropriately with proper citations.
And the manuscript is well written.
I have just several minor comments.
1. In figure 1, MHC looks like MHC class 1. In this context, MHC class II is more suitable, since adjacent T cell seems to be Th0 cells. Type of MHC should be specified, and, if necessary, the illustration of MHC should be modified.
2. In figure 1, RF can be presented as IgM, pentamers. If the authors intended to show IgG form, they can keep this illustration.
3. In table 2, Abatacept and tocilizumab can be applied as "sc" as well as IV. This can be an optional revision if the approved routes of administration vary depending on the region.
4. In Table 2, Filgotinib and Peficitinib can be added as approved JAK inhibitors.
5. TLR stands for Toll-like receptor. Please correct the typos on pages 5 (figure legend) and 17.
Author Response

(The authors gave the same response as above.)
